# CONSCIOUS INFERENCE FOR OBJECT DETECTION

## ABSTRACT

Current Convolutional Neural Network (CNN)-based object detection models adopt strictly feedforward inference to predict the final detection results. However, the widely used one-way inference is agnostic to the global image context and the interplay between input image and task semantics. In this work, we present a general technique to improve off-the-shelf CNN-based object detection models in the inference stage without re-training, architecture modification or ground-truth requirements. We propose an iterative, bottom-up and top-down inference mechanism, which is named conscious inference, as it is inspired by prevalent models for human consciousness with top-down guidance and temporal persistence. While the downstream pass accumulates category-specific evidence over time, it subsequently affects the proposal calculation and the final detection. Feature activations are updated in line with no additional memory cost. Our approach advances the state of the art using popular detection models (Faster-RCNN, YOLOv2, YOLOv3) on 2D object detection and 6D object pose estimation.

## 1 INTRODUCTION

The goal of object detection is to localize every instance from a set of predetermined categories within any given image. In recent years a large number of works have advanced visual object detection (Girshick et al. (2014); Girshick (2015); Ren et al. (2015); Redmon et al. (2016); Liu et al. (2016)), while building on the success of Convolutional Neural Networks (CNNs) as rich feature extractors. However, despite the impressive performance of the existing detectors in the datasets that they are trained on, their generalization power typically suffers when they are deployed on unseen data. The largely shifted data distribution caused by various external factors, such as camera distance, lighting conditions, background variation etc, tremendously degrades the performance. Besides, even for objects of the same class, the intra-class variability is a hard problem to handle.

Such shifted data distribution problem does not only exist in object detection, but it also appears in many other computer vision tasks including segmentation Sankaranarayanan et al. (2017) and visual question answering Wang et al. (2018). Recent approaches tackle this issue by predicting saliency (Stone et al. (2017); Sankaranarayanan et al. (2017); Wang et al. (2018)). Stone et al. (2017) require extra mask supervision during training in order to learn to mask out the irrelevant to the class of interest regions at test time. Wang et al. (2018) rely on ground-truth information either for a subset of classes or for an auxiliary task (visual question answering). Unlike these works, our method requires no additional prior information. Instead, it utilizes the models category-specific posterior confidence to highlight the pixels of positive evidence for each detected object and suppress the background. In line with our strategy, Sankaranarayanan et al. (2017) use structural perturbations from pixel-wise predictions in order to utilize context and improve semantic segmentation. **To the best of our knowledge, this work is the first online inference algorithm to address the shifted testing data distribution problem in object detection**.

Unlike classification tasks which only rely on the class-specific feedback, an equally important factor in object detection is the bounding box (bbox) estimation for each prediction. Our self-correcting algorithm is designed to guide inference via category-specific, generic object and localization evidence using corresponding losses. Other recent methods (Jiang et al. (2018); Pirinen & Sminchisescu (2018); Rao et al. (2018); Xie et al. (2018)) improve baseline two-stage object detectors by introducing an assistant network, typically instantiated as a reinforcement learning agent, which demands parameter learning during training. **Our approach is directly applicable to CNN-based**

**detectors without extra layer training, and can be used for both one-stage and two-stage frameworks.**

In this work, **we design a general online algorithm that improves off-the-shelf performance of pre-trained CNN-based detectors at inference time without re-training, architecture modification or any ground-truth requirements.** An iterative detection algorithm is proposed to utilize the positive evidence provided by the top-down feedback flow for input refinement. The updated feature activation is then fed to the *same* network again to start a new round of inference. While the downstream pass accumulates category-specific evidence over time, it subsequently affects the proposal calculation and the final detection. Since feature activations are updated in line and the network parameters are fixed, there is no additional memory cost. Our method achieves significant improvement for different state-of-the-art object detectors (Faster-RCNN(Ren et al. (2015)), YOLOv2(Tekin et al. (2018)), YOLOv3(Redmon & Farhadi (2018))) in two different computer vision tasks, that is 2D object detection and 6D object pose estimation.

Our proposed algorithm is inspired by well-founded theories for human brain consciousness of which two main aspects are captured by our algorithm, the **temporal persistence** in human perception and **top-down feedback** signal. As illustrated by the Global Neuronal Workspace theory proposed by Dehaene (2014), the consciousness is the global sharing of information within the human brain. To achieve this state of global ignition, both temporal and top-down signals are critical drives. The concept of Guided Search as an attention mechanism which is guided by the output of earlier processes has been well-established in pre-CNN literature (Wolfe et al. (1989); Tsotsos et al. (1995)). Our proposed guided, iterative inference algorithm is designed to model both temporal persistence and top-down guidance and it is therefore termed as *conscious inference*.

## 2    RELATED WORK

Our work is closely related to CNN-based object detection methods, refinement techniques for existing object detectors, self-corrective CNN techniques and other areas.

### 2.1    CNN-BASED OBJECT DETECTOR

Recently, CNN-based object detectors have achieved overwhelming success to dramatically improve the state of the art in detection. A series of region proposal-based detectors (R-CNN Girshick et al. (2014), Fast R-CNN Girshick (2015), Faster R-CNN Ren et al. (2015), etc) are designed to develop and accelerate detection by sharing CNN features and combining CNN-based Region Proposal Network (RPN), respectively. Since these detectors exhibit a two-stage propose-refine pipeline, the detection accuracy is promising but they suffer from high computation burden. Therefore, several single-shot one-stage detectors Lin et al. (2018); Liu et al. (2016); Redmon & Farhadi (2018) have been proposed which aim to achieve real-time detection by utilizing an anchor-refine pipeline. For both families of detectors, the final detection results are obtained by a bottom-up one-way inference process. However, such one-way inference is agnostic to the global image context.

### 2.2    OBJECT DETECTOR REFINEMENT

Besides designing novel object detection network architectures, several works focus on how to further improve the performance of the existing detectors with no or minor architecture modification. By modifying the standard cross-entropy gradient, Rao et al. (2018) proposed a simple yet effective method to learn globally optimized detector for object detection based on scores and bounding boxes. In that case no modification in the network architecture is needed. Pirinen & Sminchisescu (2018) proposed a deep reinforcement learning-based RPN which replaces the greedy RoI selection process with a sequential attention mechanism trained via reinforcement learning. Jiang et al. (2018) designed a side network, called IoU-Net, to predict the IoU between each detected bounding box and the matched ground-truth. The non-maximum suppression (NMS) procedure is improved by preserving accurately localized bounding boxes according to the obtained localization confidence. Cai & Vasconcelos (2018) unwrap Faster R-CNN to a sequence of detectors trained with progressively increasing IoU thresholds. The detectors are trained stage by stage, which adds parameters and training overhead linearly to the number of stages. The aforementioned detector refinement methods are not only limited by one-way inference but they also add extra training cost.

## 2.3 Self-Corrective Behavior in CNN

In recent years, works exploiting the capability of CNNs to improve their off-the-shelf performance without re-training or additional data have attracted more and more attention. Sankaranarayanan et al. (2017) proposed a self-corrective mechanism for semantic segmentation. The structural perturbation generated by computing the error between output prediction and pseudo ground-truth is injected to the input image, expecting the updated prediction from the perturbed input to be improved due to the context. Carreira et al. (2016) designed an error feedback layer to iteratively refine the human pose estimation. Such self-correction algorithm is different from ours since our proposed top-down conscious feedback is totally parameter-free without any prior training needed or any model modification. Wang et al. (2018) proposed an inference procedure to iteratively update the feature maps, targeting to improve the predictions for unknown classes when partial evidence is available. However, it uses ground-truth for a subset of known classes or for the predictions of an auxiliary task, which is a very restrictive requirement in practice during inference. Instead, our inference scheme needs no ground-truth information. Additionally, these self-correction methods only utilize the category-specific feedback signal for error correction, which is not sufficient for object detection task. For our inference method, different kinds of perturbations are generated specifically for improving both recognition and localization accuracy.

## 3 Conscious Inference For Object Detection

### 3.1 One-Way Inference For Object Detection

We start by reviewing the general one-way inference process of traditional object detectors (Fast/Faster R-CNN, YOLO, SSD, etc) briefly. Let $I \in \mathbb{R}^{W*H*C}$ represent the input image for detection, $\mathbb{B} \in \mathbb{R}^{D*k}$ is the $k$ ground-truth object bboxes. In order to train a baseline detector network with parameter $\Theta(.)$, a loss function $\mathcal{L}(\Theta(.), I, \mathbb{B})$ is usually optimized. CNN training is out of the scope of this work so we assume the network $\Theta(.)$ has already been well-trained and remains unaltered at inference stage. During testing, the one-way inference gives the $m$ detection predictions as $\mathbb{B}_{pred} = \Theta(I) \in \mathbb{R}^{D*m}$, that each prediction $B_i \in \mathbb{R}^D$ contains the bbox localization, objectness score and class score information. Finally the non-maximum suppression (NMS) is performed to obtain the final detection results $\mathbb{B}_{pred}^{nms} \subseteq \mathbb{B}_{pred} \in \mathbb{R}^{D*n}, n \leq m$. The one-way inference result $\mathbb{B}_{pred}^{nms}$ is compared with the ground-truth $\mathbb{B}$ to quantitatively evaluate the detection performance.

### 3.2 Conscious Inference (CI) For Object Detection

To facilitate the understanding, in the this section, we present our proposed conscious inference algorithm using a recent end-to-end one-stage object detector, YOLOv3 Redmon & Farhadi (2018), as a baseline. This model is a new version of the classic one-shot object detector Redmon et al. (2016). Assuming the updated input after the $(r-1)_{th}$ round of conscious inference is $I^{r-1}$, then the detection prediction of input $I^{r-1}$ is $\mathbb{B}_{pred}^r$. Each dimension of $\mathbb{B}_{pred}^r$ is represented by bbox $B^i = [b_h^i, b_w^i, h^i, w^i, o^i, c_1^i, c_2^i, ..., c_l^i] \in \mathbb{R}^D$, where $\{b_h^i, b_w^i, h^i, w^i\}$ are 4 bounding box offsets, $o^i$ is the objectness score and $\{c_1^i, c_2^i, ..., c_l^i\}$ are $l$ class prediction scores. Since the final detection results after NMS $\mathbb{B}_{pred}^{nms,r} \subseteq \mathbb{B}_{pred}^r$ are the most reliable detection predictions, they are further used to filter the ROIs [1] so that the selected ROI candidates $\mathbb{B}_{roi}^r \subseteq \mathbb{B}_{pred}^r$ have the following properties: (1) high IoU score over a threshold $\lambda_{iou}$ (they are highly overlapped with $\mathbb{B}_{pred}^{nms,r}$); (2) high objectness score over a threshold $\lambda_{obj}$ (with high probability, the selected ROI contains objects). Therefore, for one ROI $B_{roi}^i$ in $\mathbb{B}_{roi}^r$ extracted by $B^* = [b_h^*, b_w^*, h^*, w^*, o^*, c_1^*, c_2^*, ..., c_l^*] \in \mathbb{B}_{pred}^{nms,r}$, extending Sankaranarayanan et al. (2017) to form a three-fold guided signal, three pseudo ground-truths are generated respectively as:

---

[1]There is no concept of "ROI" in one-stage detectors without RPN, but we use the term "ROI" here to represent the detection predictions before NMS, which in practice are based on network stride and anchors.

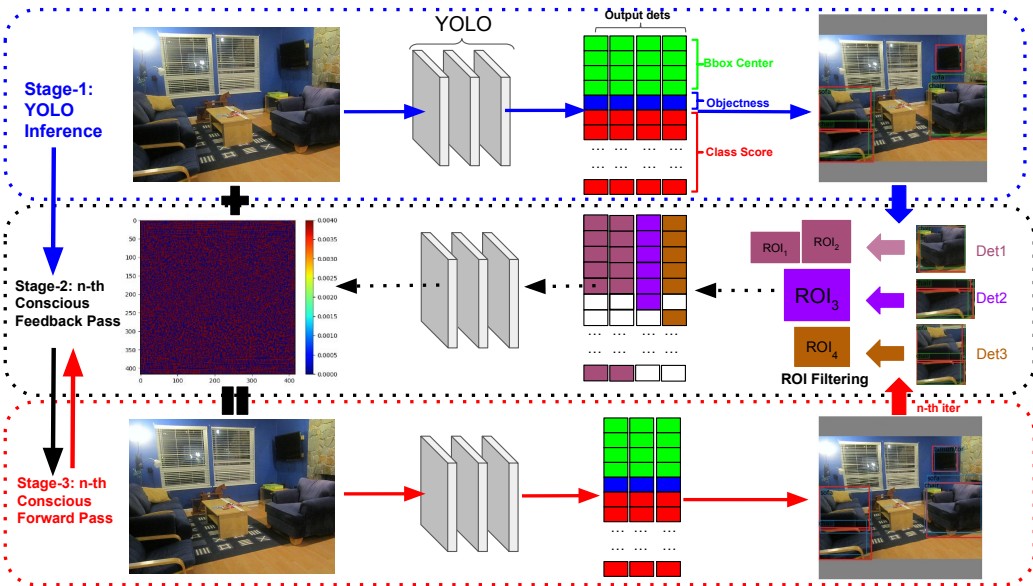

Figure 1: An illustration of the pipeline for our proposed conscious inference on YOLOv3.

$$B_{box}^{r,i} = [\underbrace{b_h^*, b_w^*, h^*, w^*}_{\text{bbox}}, \underbrace{0}_{\text{objectness}}, \underbrace{0, 0, ..., 0}_{\text{class}}]$$

$$B_{obj}^{r,i} = [\underbrace{0, 0, 0, 0}_{\text{bbox}}, \underbrace{1}_{\text{objectness}}, \underbrace{0, 0, ..., 0}_{\text{class}}] \tag{1}$$

$$B_{cls}^{r,i} = [\underbrace{0, 0, 0, 0}_{\text{bbox}}, \underbrace{0}_{\text{objectness}}, \underbrace{0, 0, ..., \underbrace{1}_{\text{i-th cls}}, .., 0}_{\text{class}}]$$

Next, the $r_{th}$ conscious feedback pass will generate three kinds of perturbations respectively as:

$$P_{box}^r = \epsilon \cdot step\left(\bigtriangledown_{I^{r-1}} \mathcal{L}(\Theta(.), I^{r-1}, \mathbb{B}_{box}^r)\right)$$

$$P_{obj}^r = \epsilon \cdot step\left(\bigtriangledown_{I^{r-1}} \mathcal{L}(\Theta(.), I^{r-1}, \mathbb{B}_{obj}^r)\right) \tag{2}$$

$$P_{cls}^r = \epsilon \cdot step\left(\bigtriangledown_{I^{r-1}} \mathcal{L}(\Theta(.), I^{r-1}, \mathbb{B}_{cls}^r)\right)$$

where $\epsilon$ is the weighting parameter, $step(.)$ is the step activation function, $\bigtriangledown_{I^{r-1}}\mathcal{L}()$ is the gradient map w.r.t input layer, $P_{box}^r$, $P_{obj}^r$ and $P_{cls}^r$ are the bounding box, objectness and class-specific perturbations, respectively. Based on our experimental observation, $step(.)$ performs better than the $signum$ activation function used in Sankaranarayanan et al. (2017).

By updating $I^{r-1}$ as $I^r = I^{r-1} + (P_{boc}^r + P_{obj}^r + P_{cls}^r)$, the $(r+1)_{th}$ conscious inference round is performed:

$$\mathbb{B}_{pred}^{r+1} = \Theta(I^r) \tag{3}$$

The overall conscious inference process is illustrated in Fig. 1. As we claimed, our proposed conscious inference algorithm is a general scheme that is applicable to any existing CNN-based object detectors. There is no requirement for global average pooling layer as in Zhou et al. (2016). We next briefly describe how to implement our method to other baseline detection-based networks.

**CI For Two-Stage Detector**: The conscious inference algorithm is also implemented on top of a classic two-stage object detector, Faster R-CNN Ren et al. (2015). By design of the RPN, perturbations $P_{obj}^r$ and $P_{box}^r$ are generated from the $RPN_{cls}$ and $RPN_{bbox}$ layers and added to the $RPN_{conv1}$.

As for the $P_{cls}^r$, the $RCNN_{cls}$ layer predicts the class score of each $B^i$ from $RCNN_{bbox}$. Thus, following the same way used for YOLOv3, a perturbation $P_{cls}^r$ can be obtained and added to the base feature map (the output of the backbone network). Compared with the implementation on YOLOv3, our Faster R-CNN conscious inference version is more efficient since the conscious feedback pass goes less deep.

**CI For Pose Estimator**: A state of the art object pose estimator Tekin et al. (2018) is integrated to benchmark conscious inference in a scenario that naturally extends the $2D$ detection paradigm to a richer $6D$ representation. A YOLOv2-like end-to-end network with a modified regression output layer and a PnP algorithm to fit the pose estimation requirement is proposed in Tekin et al. (2018) to jointly detect the object as well as estimate its pose. Similar as in Eqn. 2, $P_{box}^r$, $P_{obj}^r$ and $P_{cls}^r$ can be readily generated following the same manner.

## 4    EXPERIMENTS

We validate the effectiveness and efficiency of conscious inference on different object detection baselines and tasks: Sec. 4.1 shows the results of the one-stage detector YOLOv3, Sec. 4.2 presents the results of the two-stage detector Faster R-CNN and Sec. 4.3 demonstrates the performance on a 6D pose estimator. All the experiments are conducted on a single NVIDIA TITAN Xp GPU.

### 4.1    EXPERIMENTS ON YOLOV3 FOR 2D OBJECT DETECTION

**Dataset and Evaluation.**  The experiment is conducted on MS-COCO Lin et al. (2014), where the 5k subset of validation (*minival*) images are tested. For evaluation, the standard COCO-style Average Precision (AP) across IoU thresholds from 0.5 to 0.95 with an interval of 0.05 is adopted.

**Experimental Setting.** The pre-trained YOLOv3 model from Redmon & Farhadi (2018) and the exact same algorithm parameter setting are adopted in our experiments without any modification. For our conscious inference algorithm, we set $\epsilon = 0.004$, $\lambda_{obj} = 0.5$ and $\lambda_{iou} = 0.9$. Unless otherwise stated, the result from the first conscious iteration is reported due to most favorable performance and time efficiency trade-off. Results from more iteration rounds are also reported in Table 1 and Fig. 4.

**Experimental Results and Ablation Study.** The results on COCO *minival* are reported in Table 1. Largest improvement is achieved after one iteration, while more inference rounds have diminishing returns. Several ablation studies are also conducted. (1) The influence from different perturbations can be compared in Table 2. As it can be seen, $P_{box}$ performs well for refining the bbox of originally detected object (improvement on high IoU); $P_{obj}$ is able to discover more missing objects (large improvement on AP); $P_{cls}$ focuses more on correcting the wrong classification of existing detections. (2) Some parameter search experiments are conducted and shown in Fig. 3. With the increase of $\epsilon$, the performance keeps raising at first then turn to decrease since a large $\epsilon$ will over-modify the original input. Larger $\lambda_{iou}$ is able to filter more reliable and confident ROI candidates so better performance is achieved. If we keep raising $\lambda_{obj}$, some reliable ROI candidates will be mistakenly eliminated causing a worse result. (3) Different feedback perturbation strategies are compared in Fig. 4. Besides the quantitative results, some visualization results are presented in Fig. 2 and Fig. 5.

| Models | AP | AP$_{50}$ | AP$_{75}$ |
|---|---|---|---|
| YOLOv3 | 39.67 | 58.28 | 45.00 |
| Our-iter1 | 40.11 | 58.93 | 45.45 |
| Our-iter2 | 40.11 | 59.14 | 45.43 |
| Our-iter3 | 40.13 | 59.23 | 45.53 |
| Our-iter4 | 40.14 | 59.30 | 45.53 |
| Our-iter5 | 40.13 | 59.36 | 45.46 |

Table 1: Comparison between conscious inference against standard inference on YOLOv3 across iterations.

| Models | AP | AP$_{50}$ | AP$_{75}$ | AP$_{95}$ |
|---|---|---|---|---|
| YOLOv3 | 39.67 | 58.28 | 45.00 | 1.60 |
| YOLOv3+OP | 40.01 | 58.87 | 45.36 | 1.53 |
| YOLOv3+BP | 39.65 | 58.26 | 44.96 | 1.63 |
| YOLOv3+CP | 39.87 | 58.65 | 45.21 | 1.58 |
| YOLOv3+OP+BP | 40.02 | 58.87 | 45.39 | 1.56 |
| YOLOv3+OP+CP | 40.08 | 58.93 | 45.41 | 1.59 |
| YOLOv3+CP+BP | 39.85 | 58.62 | 45.21 | 1.60 |
| YOLOv3+OP+CP+BP | 40.11 | 58.93 | 45.45 | 1.61 |

Table 2:   The influence of different perturbations. BP=$P_{box}$, OP=$P_{obj}$ and CP=$P_{cls}$. Iter-1 result is reported.

| Models | YOLOv3 | Our-Input | Our-conv1 | Our-conv12 | Our-conv35 | Our-conv81 |
|--------|--------|-----------|-----------|------------|------------|------------|
| AP | 39.67 | 40.11 | 39.99 | 39.69 | 39.67 | 39.65 |
| $AP_{50}$ | 58.28 | 58.93 | 58.77 | 58.40 | 58.28 | 58.26 |
| FPS | 33 | 4 | 5 | 7 | 11 | 15 |

Table 3: The trade-off between efficiency and effectiveness of conscious inference on YOLOv3.

Table 3 shows the trade-off between efficiency and effectiveness of our algorithm by truncating the back-propagation pass and injecting the generated perturbations into different layers. The deeper the feedback signal goes, the better the improvement is while the computation increases. The time overhead is linear to the number of layers that the algorithm traverses during sequential top-down and bottom-up passes. Next we show an efficient implementation of our method for Faster R-CNN.

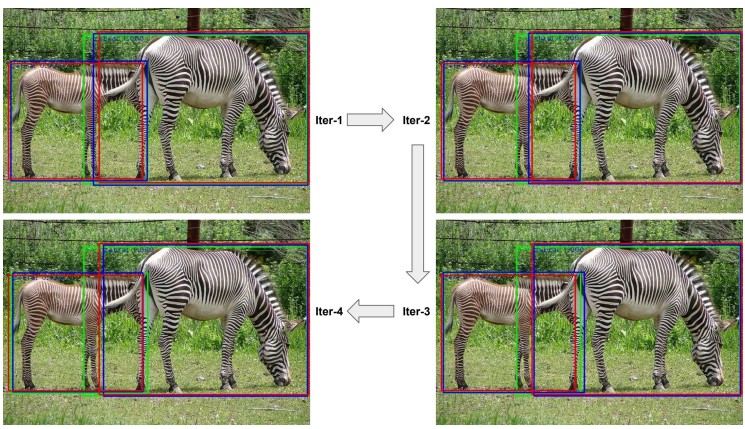

Figure 2: The visualization of consistent improvement of detection bbox in a sample image. Red bbox is the ground-truth, green bbox is the baseline result and blue bbox is our conscious inference result.

## 4.2 EXPERIMENTS ON FASTER R-CNN FOR 2D OBJECT DETECTION

**Dataset and Evaluation.** Besides the MS-COCO *minival* evaluated in Sec. 4.1, PASCAL VOC 2007 Everingham et al. (2010) dataset is also tested. There are 5k trainval images and 5k test images over 20 object categories. Compared with MS-COCO which often contains multiple small objects in one image, PASCAL VOC 2007 focuses more on large objects, so the accuracy of predicted bounding box is even more crucial here. Like COCO-style evaluation metric, the mean Average Precision (mAP) over different IoU thresholds is tested.

**Experimental Setting.** We use a pre-trained Faster R-CNN model (Res101 backbone) from a pytorch implementation [2], which achieves comparable performance against (Ren et al., 2015). For our

[2]https://github.com/jwyang/faster-rcnn.pytorch

| Model | AP | $AP_{50}$ | $AP_{75}$ | $AP_S$ | $AP_M$ | $AP_L$ | $AR_S$ | $AR_M$ | $AR_L$ |
|-------|-----|-----|-----|------|------|------|------|------|------|
| Faster (Res101) | 34.5 | 54.9 | 36.9 | 14.4 | 39.2 | 52.4 | 22.8 | 52.1 | 66.4 |
| Faster+OP | 34.6 | 55.0 | 36.8 | 14.4 | 39.3 | 52.4 | 22.8 | 52.3 | 66.5 |
| Faster+BP | 34.5 | 54.9 | 37.0 | 14.5 | 39.3 | 52.5 | 22.9 | 52.3 | 66.5 |
| Faster+CP | 34.5 | 54.9 | 36.9 | 14.5 | 39.2 | 52.5 | 22.9 | 52.1 | 66.4 |
| Faster+OP+BP | 34.7 | 55.0 | 37.1 | 14.6 | 39.5 | 52.8 | 23.0 | 52.5 | 67.1 |
| Faster+OP+BP+CP | 34.7 | 55.0 | 37.1 | 14.5 | 39.4 | 52.8 | 22.9 | 52.4 | 66.9 |

Table 4: The influence of different perturbations on COCO. Iter-1 result is reported.

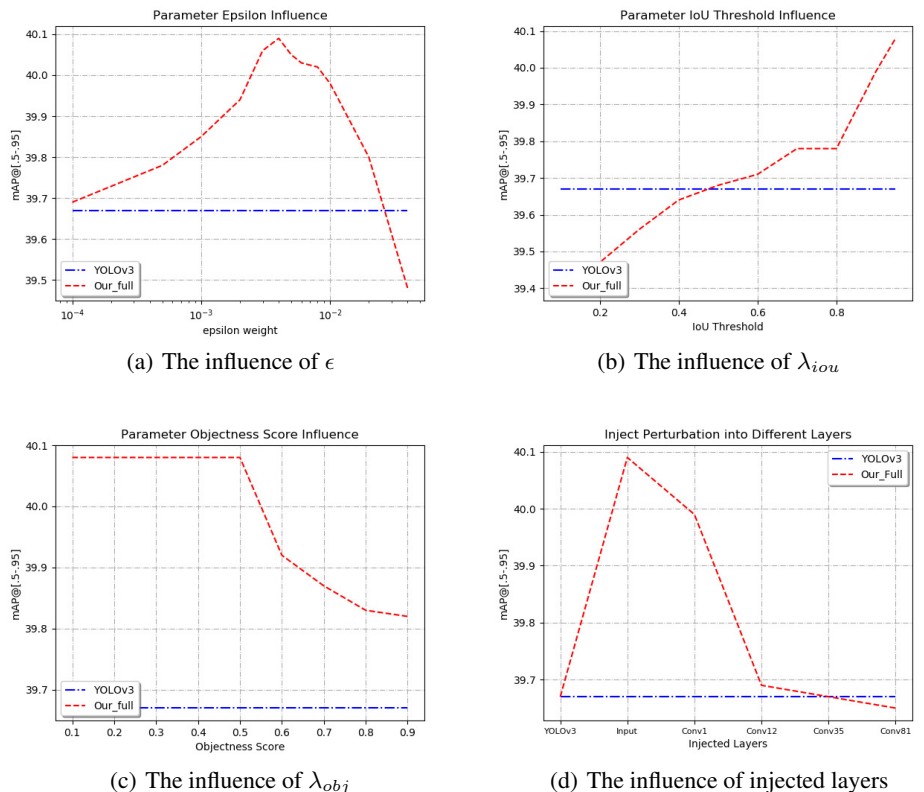

Figure 3: Ablation studies for various factors. **(a)**: The influence of $\epsilon$. **(b)**: The influence of $\lambda_{iou}$. **(c)**: The influence of $\lambda_{obj}$. **(d)**: The influence of injecting perturbation into different layers.

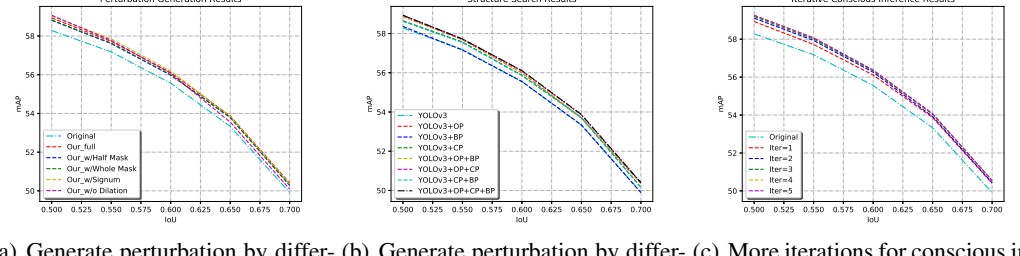

(a) Generate perturbation by different components

(b) Generate perturbation by different structure designs

(c) More iterations for conscious inference

Figure 4: **(a)**: Different strategies to generate combined perturbation $P_{boc}^r + P_{obj}^r + P_{cls}^r$. **(b)**: Ablation study on using different components. **(c)**: Detection performance across inference rounds.

| Model | AP | $AP_{50}$ | $AP_{60}$ | $AP_{70}$ | $AP_{80}$ | $AP_{90}$ |
|---|---|---|---|---|---|---|
| Faster (Res101) | 46.1 | 74.1 | 66.2 | 52.5 | 30.8 | 6.9 |
| Faster+OP+BP+CP | 46.8 | 74.3 | 66.4 | 53.3 | 31.4 | 8.4 |

Table 5: The results on PASCAL VOC 2007. Iter-1 result is reported.

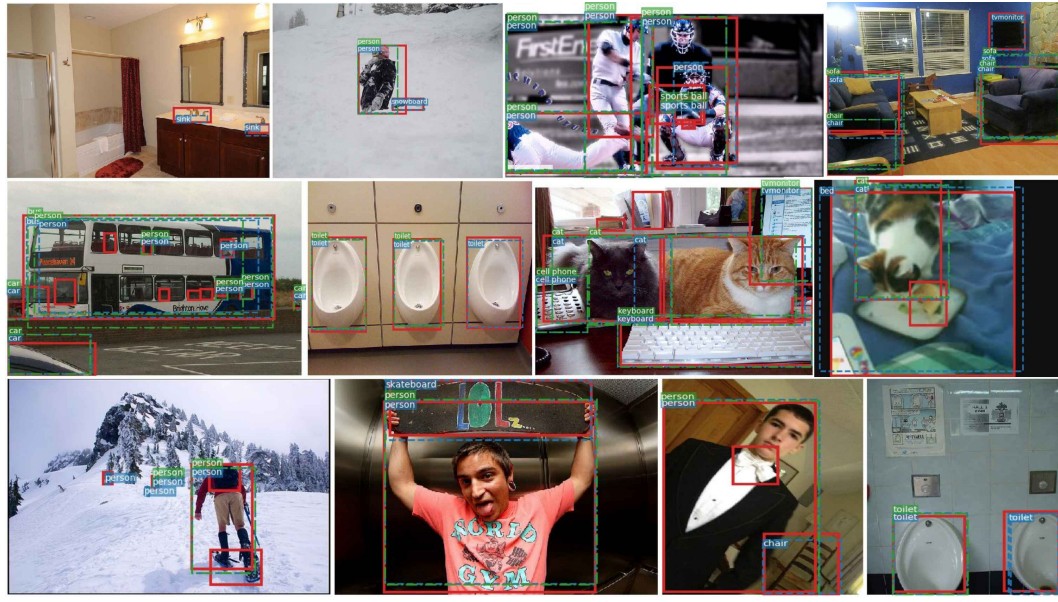

Figure 5: The visualization of detection improvement on COCO minival based on YOLOv3 detector. The red bbox is the ground-truth, green bbox is the detection result from baseline (one-way inference) and blue bbox is the detection result from our conscious inference (iter-1).

inference algorithm, we set $\epsilon = 0.020$, $\lambda_{obj} = 0.9$ and $\lambda_{iou} = 0.4$. Unless otherwise stated, the result from the first iteration is reported.

**Experimental Results.** Tables 4 and 5 and Fig. 6 report the detection results on COCO and PAS-CAL VOC 2007 datasets respectively. Conscious inference improves the performance under various evaluation criteria, especially for high IoU and large objects. However, the gain is not as large as with YOLOv3, which can be partially attributed to the shallower conscious feedback pass.

### 4.3 EXPERIMENTS ON YOLOV2-6D FOR 6D OBJECT POSE PREDICTION

**Dataset and Evaluation.** The extremely challenging multi-object detection and pose estimation dataset, OCCLUSION Brachmann et al. (2014), is evaluated in our experiment. As its name suggests, several objects in the images are severely occluded due to scene clutter, which makes pose

| Category | Ape | Can | Cat | Driller | Glue | Holepuncher |
|---|---|---|---|---|---|---|
| $Acc_{5px}$ | 6.07\7.01 | 10.11\11.35 | 3.45\3.45 | 1.07\1.24 | 5.20\5.32 | 8.10\9.50 |
| $Acc_{10px}$ | 39.32\43.25 | 58.16\58.99 | 21.74\21.74 | 16.97\16.97 | 25.69\25.91 | 38.84\38.93 |
| $Acc_{15px}$ | 59.83\63.08 | 79.70\79.29 | 38.42\38.50 | 40.44\40.77 | 39.20\39.09 | 52.73\52.81 |
| $Acc_{20px}$ | 68.29\71.11 | 86.00\85.75 | 49.20\49.20 | 62.11\62.27 | 46.73\46.84 | 62.40\62.23 |
| $Acc_{25px}$ | 72.74\74.87 | 88.48\88.24 | 54.42\54.59 | 80.15\80.40 | 50.06\51.16 | 68.84\68.84 |
| $Acc_{30px}$ | 74.96\77.26 | 90.89\90.47 | 58.05\58.13 | 89.95\89.99 | 52.49\53.27 | 73.97\73.97 |
| $Acc_{35px}$ | 75.31\78.29 | 91.88\91.71 | 59.98\60.07 | 93.90\94.15 | 53.71\54.60 | 80.17\80.25 |
| $Acc_{40px}$ | 76.32\78.72 | 92.79\92.46 | 60.99\61.08 | 95.47\95.47 | 54.15\55.26 | 85.45\85.62 |
| $Acc_{45px}$ | 76.67\79.15 | 93.45\93.04 | 62.34\62.43 | 96.29\96.62 | 54.82\55.81 | 89.50\89.50 |
| $Acc_{50px}$ | 78.03\79.49 | 93.79\93.37 | 63.27\63.27 | 96.71\96.87 | 55.26\56.04 | 91.90\91.90 |
| $\mathbf{Acc}_{mean}$ | **62.75\65.22** | **78.52\78.47** | **47.18\47.25** | **67.30\67.48** | **43.73\44.33** | **65.19\65.36** |

Table 6: Comparison of our conscious inference against standard inference using the 6D pose prediction baseline Tekin et al. (2018). As customary, 2D reprojection error is the evaluation metric. Results are shown in format **baseline\ours**, where blue means improvement and red means decline.

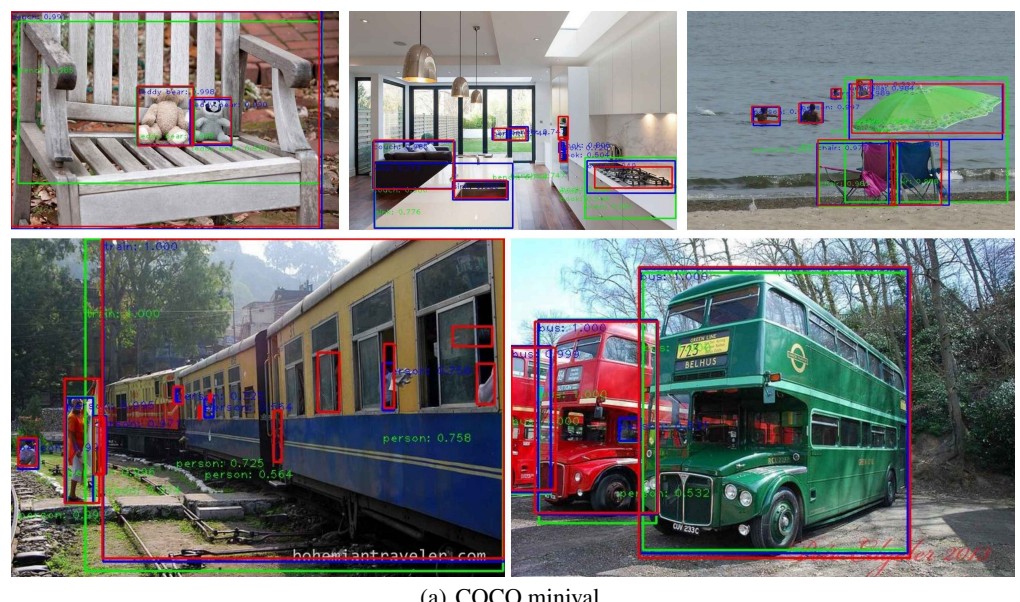

(a) COCO minival

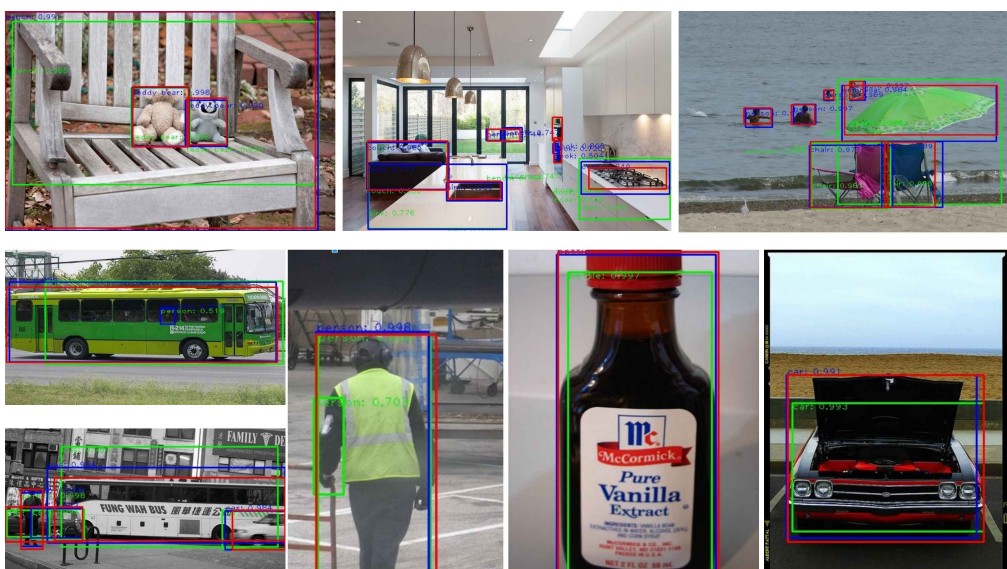

(b) VOC 2007

Figure 6: The visualization of detection improvement on (a) COCO minival and (b) VOC 2007 datasets based on Faster R-CNN detector. The red bbox is the ground-truth, green bbox is the detection result from baseline (one-way inference) and blue bbox is the detection result from our conscious inference (iter-1).

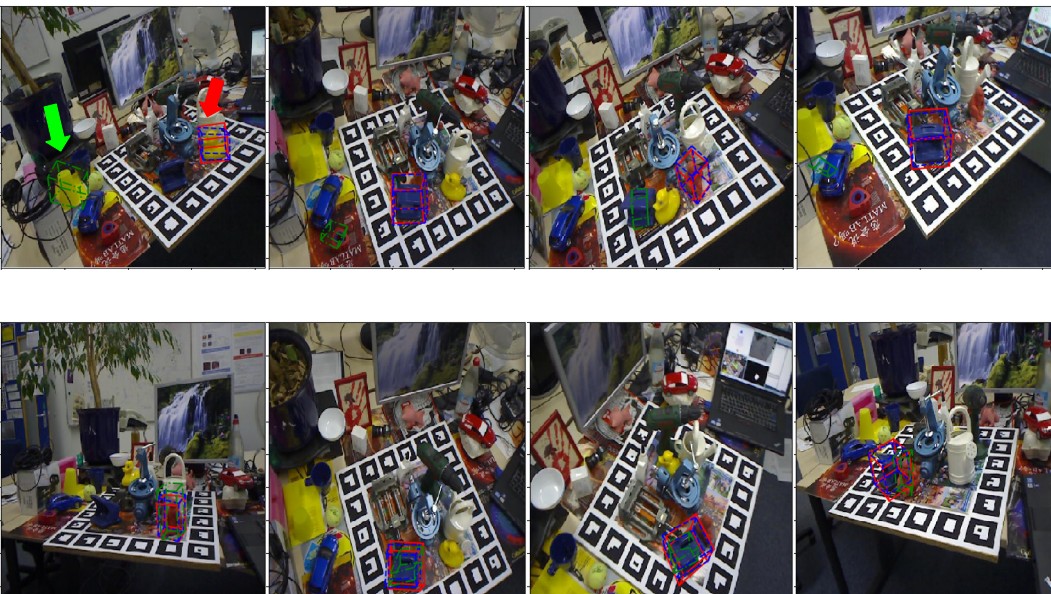

Figure 7: The visualization of pose refinement on OCCLUSION dataset based on Tekin et al. (2018). The red bbox is the ground-truth, green bbox is the detection result from baseline (one-way inference) and blue bbox is the detection result from our conscious inference (iter-1).

estimation extremely challenging. The widely-used 2D reprojection error Brachmann et al. (2016) under various thresholds is adopted to evaluate 6D pose estimation accuracy.

**Experimental Setting.** The pre-trained YOLOv2-6D pose estimation model from Tekin et al. (2018) is used. For all the 13 categories, we follow the same experimental setting as in Tekin et al. (2018), where 7 of them (ape, can, cat, driller, duck, glue, holepuncher) are used for testing.

**Experimental Results.** The comparison results are shown in Table 6. For each grid cell in the table, the first number is the baseline result and the second number is ours. As can be seen, for the most strict evaluation criterion, $\text{Acc}_{5px}$, our conscious inference is able to improve the baseline performance by a large margin. The overall performance of all categories is improved. Some visualization improvement results are shown in Fig. 7.

## 5 CONCLUSION

In this paper, we propose a guided, iterative inference algorithm, which can be applied on general CNN-based object detectors at inference stage. The proposed approach does not involve any model modification, re-training or ground-truth requirements. The term "conscious" is inspired by Dehaene (2014) since our method models two important traits identified in human cognition: top-down feedback and temporal persistence. Experiments based on different state-of-the-art object detectors show consistent improvement in diverse detection tasks. Our algorithm has no memory overhead (as opposed to strong detection refinement frameworks Cai & Vasconcelos (2018)), while the added computation is linear to the number of traversed layers. The trade-off between extra computation and performance gain is an important factor to consider based on the application scenario. Empirical results on this trade-off using our straight-forward implementation are presented in Table 3.

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
