# OpenReview forum: "Conscious Inference for Object Detection"
_ICLR.cc/2019/Conference_

### Official Review · AnonReviewer1 · 2018-10-31
**Paper review**

**Rating:** 4
**Confidence:** 5

**Review:**

The paper proposes a iterative approach at inference time to improve object detections. The work relies on updating the feature activations and perform new feed forward passes to obtain improved results.

Pros:
(+) The idea of iterative inference is potentially effective
(+) The paper is well written and clear
(+) The authors show results on compelling benchmarks
Cons:
(-) Reported improvements are very small
(-) Important baselines are missing


First, while the authors state correctly that their updates have no memory cost and no new parameters are added, they do require more FLOPs at test time. For N-stages, the approach requires xN more operations for forward passes  and xN for backward passes. This is a serious shortcoming as it adds compute time per image for the inference stage and cannot be parallelized.

The authors show small improvements for AP on COCO. From their analysis, it seems that the biggest gains come from N=1 stages, while the improvement added for N>1 are miniscule (Table 1). Note that the authors show results on COCO minival (5k images) and from my experience there, it's expected to see a +/- 0.2% AP between different trained models of the same architecture. The authors report a +0.46% gain.

In addition, the authors do not provide results for other baseline approaches that have similar FLOPs at test time, such as iterative bounding box regression and input scale augmentation. Note that both these approaches do not add any parameters and require no additional memory, but add to the FLOPs at test time. From my personal experience, test time augmentations can add +1.5% to the final performance. Concretely, look at Mask R-CNN arXiv Table 8 last two rows. Test time augmentations add 1.5% on top of an already enhanced model. Empirically, the better the model the harder it is to get gains from inference tricks! And still test time augmentations boost performance significantly.

Given the small gains and the lack of competing baselines, it is hard to make a case for accepting the paper.

---

> ### Author Response · Authors · 2018-11-23
> **Thanks a lot for your review!**
>
> Thank you very much for your time and effort for reviewing our paper. We will keep working on refining our work according to your precise comments and suggestions.

---

### Official Review · AnonReviewer3 · 2018-11-02
**The proposed approach is overall interesting but the overall gain seems a bit small given the iterative nature of the method that slows inference quite a bit down.**

**Rating:** 6
**Confidence:** 4

**Review:**

The paper proposes a method called cautious inference to improve inference accuracy for object detection models. The main idea is inspired by the previous work of Guided Perturbations, which is applied to fully convolutional networks to improve the segmentation/accuracy accuracy purely during inference time.  The original idea is to use the predicted labels of the network as pseudo ground truths (after making the predictions to be a one-hot vector), and then back propagate the error signals to the network input to get the gradients. And finally the gradients are added back to the original inputs to perform another round of prediction. Here the inputs can be either the original image, or some intermediate feature maps. Experiments are shown for both 2D and 6D object detections.

Comments:

- I think overall it is an interesting idea to directly alter the input of the network in order to fit to the testing distribution. However, the motivation and story told in the introduction is a bit of an oversell compared to the experiment validation section. Most of the results shown are just doing training and testing of images drawn from the *same* distribution. Like coco train and test, or VOC train and test. It would be great to see if the cautious inference would work when the distribution is different. For example "elephant in the room" case, or new object categories are added during testing.

- I am actually curious to see this method can be used to improve the AP on the *training* set as well, just to understand it better -- is it trying to recover the generalization error of the network, or it is doing some implicit context reasoning inference that can help training as well.

- It might be better to compare/combine the method to other inference-only improvements for object detection. For example there is soft-NMS,
Bodla, Navaneeth, et al. "Soft-nms—improving object detection with one line of code." Computer Vision (ICCV), 2017 IEEE International Conference on. IEEE, 2017.

 - I am not sure I fully understand B-box part: I think it is easy to have B-obj and B-cls as one can just take the max of the class prediction and then use the inferred class label for one-hot vector construction, but I am confused about the box part as no ground-truth is given during testing. In Table 2 I also cannot find BP improving performance by itself in anyway.

- For COCO, please report results on test-dev set, the minival set images are used only for validation.

---

> ### Author Response · Authors · 2018-11-23
> **Thanks a lot for your review!**
>
> Thank you very much for your time and effort for reviewing our paper. We will keep working on refining our work according to your precise comments and suggestions.

---

### Official Review · AnonReviewer2 · 2018-11-06
**The authors propose a method to improve object detection accuracy at inference time without re-training, changing network architecture, and working for both one-shot and two-stages detectors.**

**Rating:** 4
**Confidence:** 4

**Review:**

The goal of the paper clearly motivated and well described. However, the notations and figures are more complicated than necessary; hence, it is a bit hard to follow the paper in detail. There are also some missing related works about domain adaptation for object detectors. For instance,
Chen et al. "Domain Adaptive Faster R-CNN for Object Detection in the Wild" In CVPR 2018.
Inoue et al. "Cross-Domain Weakly-Supervised Object Detection through Progressive Domain Adaptation" In CVPR 2018.
The authors should cite these papers and compare with their performance.
Finally, the proposed method doesn't consistently improve the detection accuracy.
The proposed method also slows down the frame rate of the detector due to multiple iterations of feedforward/feedback inferences.

---

> ### Author Response · Authors · 2018-11-23
> **Thanks a lot for your review!**
>
> Thank you very much for your time and effort for reviewing our paper. We will keep working on refining our work according to your precise comments and suggestions.

---

### Meta-Review · Area_Chair1 · 2018-12-14
**decision**

**Confidence:** 4
**Recommendation:** Reject

**Metareview:**

The paper presents an interesting idea, but there are significant concerns about the presentation issues and experimental results (e.g., comparisons with baselines). Overall, it is not ready for publication.